# Population and Conservation Status of Bighorn Sheep in the State of Baja California, Mexico

**DOI:** 10.3390/ani14030504

**Published:** 2024-02-03

**Authors:** Guillermo Romero-Figueroa, Enrique de Jesús Ruiz-Mondragón, Eahsan Shahriary, Carlos Yee-Romero, Aldo Antonio Guevara-Carrizales, Rafael Paredes-Montesinos, Jesús Miguel Corrales-Sauceda, Israel Guerrero-Cárdenas, Raul Valdez

**Affiliations:** 1Facultad de Ciencias, Universidad Autónoma de Baja California, Ensenada 22860, Mexico; ruize56@uabc.edu.mx (E.d.J.R.-M.); carlos.yee@uabc.edu.mx (C.Y.-R.); aldo.guevara@uabc.edu.mx (A.A.G.-C.); rparedes90@uabc.edu.mx (R.P.-M.); miguel.corrales@uabc.edu.mx (J.M.C.-S.); 2Fundación Universidad Autónoma de Baja California, Asociación Civil, Mexicali 21100, Mexico; 3School of Public Health, University of California, Berkeley, CA 94704, USA; eshahriary@yahoo.fr; 4Centro de Investigaciones Biológicas del Noroeste, La Paz 23096, Mexico; guerrero04@cibnor.mx; 5Department of Fish, Wildlife and Conservation Ecology, New Mexico State University, Las Cruces, NM 88046, USA; rvaldez@nmsu.edu

**Keywords:** aerial survey, desert sheep, *Ovis canadensis*, population, status, wildlife monitoring

## Abstract

**Simple Summary:**

The bighorn sheep is a species of great ecological, cultural, economic, and social importance in Mexico. However, the current sheep population status in the state of Baja California has been unknown since 2010. The objective of this research was to update information on the abundance, distribution, and population structure of bighorn sheep in Baja California through an aerial survey. The flight was conducted in 2021, and 456 bighorn sheep were observed, resulting in an estimated 1697 ± 80 animals. The observation rate was 16 sheep recorded per hour of flight, and the ram:ewe:lamb ratio was 62:100:19. A statistical comparisons of these results with previous those of aerial surveys, which began in 1992, indicate that the Baja California bighorn sheep population has been stable for twenty-nine years, in contrast to other areas of the species’ range, where the sheep population increased during the same period. Given these results, Baja California authorities should consider modifying the current bighorn sheep conservation strategy to increase the species’ population in the state.

**Abstract:**

The bighorn sheep in Mexico is classified as at-risk by the Mexican federal government. In the state of Baja California, wild sheep can be observed throughout the length of the state from the USA–Mexico border south to the Agua de Soda mountain range. This research aimed to document the historical trend of the bighorn population based on aerial surveys conducted in 1992, 1995, 1999, 2010, and 2021, and the abundance, distribution, and structure of bighorn sheep populations in Baja California, based on an aerial survey conducted from 8–14 November 2021, covering thirteen mountain ranges. The estimated sheep population in 2021 was based on the number of individuals observed; the sightability of the animals; the area sampled; and the total area of habitat available. In 30.5 flight hours, 456 bighorn sheep were observed, with an estimated population of 1697 ± 80 individuals. The observation rate was 16 sheep sighted per hour of flight, and the ram:ewe:lamb ratio was 62:100:19. When the results of the 2021 flight were compared to the results of the previous aerial surveys, there was a large variation between the data, which was related to the lack of consistency between the sampling designs used in each study. Nevertheless, a statistical test of the results of aerial surveys conducted in the state suggest that the Baja California bighorn sheep population remained stable between 1992 and 2021. This study highlights the need to standardize wild sheep aerial surveys by defining flight paths and establishing a consistent duration of flights. On the other hand, Baja California authorities should consider modifying the current conservation strategy for bighorn sheep to increase the species’ population in the state by initiating community-based wildlife conservation programs in rural communities.

## 1. Introduction

In Mexico, the bighorn sheep (*Ovis canadensis*) is an iconic species distributed throughout the length of the Baja California Peninsula and in the state of Sonora, and these sheep have been re-established in the states of Chihuahua and Coahuila, where they were extirpated in the twentieth century [1]. Wild sheep are economically important; sport hunting permits in Mexico sell for prices ranging from USD 45,000 to over USD 100,000 [2,3].

The bighorn sheep in Mexico is classified as at-risk by the federal government [4] because populations are threatened by human-induced habitat alterations that affect their viability [5]. In the state of Baja California, the official conservation strategy for the bighorn sheep since 1990 consists solely of the prohibition of sport hunting [6]. Nevertheless, in the official state document, *State strategy for the conservation and sustainable management of the bighorn sheep (Ovis canadensis cremnobates) in Baja California,* sport hunting of bighorn sheep is contemplated as a management option to “ensure the conservation of the species *Ovis canadensis cremnobates*”, which is the overall objective of the state strategy [7].

Monitoring is an effective mechanism for generating systematic information and data to evaluate the current status, determine population trends, and to detect deleterious anthropogenic effects on the population, as well as the scope of management, conservation, and restoration actions undertaken to protect it [8,9]. The three attributes of the bighorn sheep population that must be known to guide management programs for the species are current abundance, abundance trends, and population structure. Abundance indicates the number of animals in a given area, and by comparing this information to that obtained under previous monitoring, it is possible to determine the trend of the population, i.e., whether it is stable, increasing, or decreasing. One of the primary objectives of ungulate surveys is to determine the species’ population structure: ratios of lamb to adult ewe and yearling to adult ewe. The lambing ratio or lambing rate is an important indicator of population recruitment and effective reproduction because this parameter measures the birth rate and early survivorship of lambs. The yearling ratio is important in population assessments because it is a measure of the number of lambs that reach the age of 1 year and contribute to population growth. Ratios of lamb to ewe are of limited use in predicting recruitment to a population. In contrast, ratios of yearling to ewe are better estimators but have a wide margin of error because of the difficulty of differentiating between ewes and yearlings at long distances. Complete counts are the most reliable method for determining changes in population size [10]. Yearling and adult survival are the best indicators of population viability. Although immature animals contribute most to population growth, it is the variation in adult female survival that is most critical in determining the population growth rate [11,12,13]. On the other hand, the sex ratio is a parameter that must be carefully monitored in populations exploited by sport hunting, since the decrease in the proportion of males is mainly related to overharvesting and, therefore, when it is detected, measures must be taken to reduce the pressure on populations [14]. Unstable age structures and sex ratios can have genetic or reproductive consequences.

Monitoring desert bighorn sheep populations requires significant commitments of time, money, and personnel [15], since their habitat is vast, remote, and rugged, with extreme climates, a low population density, and highly clustered distribution patterns [16]. Aerial surveys are an efficient means of studying wildlife populations, covering large expanses of territory in short periods of time [17], and are therefore considered an efficient method to estimate the abundance of bighorn sheep.

The spatial distribution of bighorn sheep in the state of Baja California have been documented by Leopold [18], Monson [19], and Lee et al. [20]. Based on this information, the habitat of the species has been divided into mountain ranges and mountain range complexes, and there has been speculation about the movement of bighorn sheep between these areas. In this regard, some authors suggest that there are three metapopulations of bighorn sheep in Baja California, each inhabiting a mountain complex composed of several mountain ranges [20,21]. Potential distribution models of the species show that the state’s mountain ranges are connected by corridors that can be used by bighorn sheep [22,23], and it is known that these animals move from one mountain range to another in response to adverse habitat conditions [24]. However, there is no evidence to determine the number of metapopulations in Baja California or whether they are divided by mountain complexes or ranges, although Buchalski’s [25] study suggests that a single range may support more than one metapopulation.

Four aerial surveys have been conducted in Baja California to assess the abundance and population structure of bighorn sheep in 1992, 1995, 1999, and 2010. The 1992, 1995, and 1999 surveys covered half of Baja California’s mountain ranges where the species occurs [26,27,28]; only the 2010 survey covered all of the state’s mountain ranges where bighorn sheep occur [29]. The aims of this study were to estimate the current abundance and population structure of bighorn sheep populations in the state of Baja California based on the aerial survey conducted in 2010 and to compare the 2021 data with those of previous surveys to establish historical population size trends.

## 2. Materials and Methods

### 2.1. Study Area

The study area encompassed thirteen mountain ranges inhabited by wild sheep in the state of Baja California [20]: Cucapá, Sierra Juárez, Las Tinajas, Las Pintas, San Pedro Mártir, San Felipe, Santa Isabel, San Francisquito, Calamajué, La Asamblea, La Libertad, Las Ánimas, and Agua de Soda (Figure 1). Together, these mountain ranges cover a surface area of 967,910.33 ha [30], extending from La Rumorosa on the Mexico–United States border south to 28°50′, approximately 50 km north of the state border with Baja California Sur.

The physiographic features of the surveyed area include high, steep-sloped mountain ranges, complex mountain ranges with plateaus, complex low mountain ranges, and dissected plateaus with valleys; basalt plateaus with knolls; and complex knolls with bajadas [30]. The dominant vegetation types are chaparral, microphyllous desert scrub, rosettophyllous desert scrub, sarco-crasicaule scrub, and sarcocaule scrub [31].

### 2.2. Aerial Surveys

We conducted an aerial survey from 8–14 November 2021, using a five-seat Bell 505 helicopter. The monitoring team consisted of three observers and the pilot, who served as a fourth observer, all with experience in aerial bighorn sheep monitoring in Mexico. The aircraft traveled at an average speed of 100 km/h and at an elevation of between 15 and 30 m from the ground [26,27,28,29]. The flight path followed the outlines of the mountain ranges, and each flight path was made from the foothills to the summit of the mountains where the altitude gain was in 500 m intervals [27]. When a sheep or group of sheep was observed, the helicopter maneuvered to fly closer to facilitate identification. The helicopter route and the location of bighorn sheep sightings were recorded using a Garmin 64 s GPS. The sheep observed were classified into age class and sex based on the criteria of Geist [32]. In this classification, sheep are divided into eight categories based on the size and shape of their horns and body size: lamb, yearling ewe, ewe, yearling ram, class I ram, class II ram, class III ram, and class IV ram (Table 1).

Aerial surveys in Baja California have not been standardized; there is no defined flight route, and the hours of flight time that must be dedicated to each mountain range have not been established. The previous aerial surveys conducted in Baja California, except for the 1992 survey [27], do not describe the flight procedures used [26,28,29], and the information on the routes followed by those flights are unavailable. For this reason, in this survey, the flight path was defined based on the experience of the monitoring team and was directed to the areas where there was a greater probability of encountering the animals.

### 2.3. Population Estimation

Important factors in determining a population estimate based on an aerial survey include the number of animals observed, the sightability value, the area sampled, and the total area available for the species. Sightability refers to the likelihood of observing an animal within the observers’ field of view [33]. During the previously conducted surveys prior to the 2021 survey, it was assumed that only between 35% and 60% of the total sheep populations were sighted [27,29]. This is based on the results of McQuivey [34] and Hervert et al. [35], who reported that the probability of detecting a group of sheep during a helicopter aerial survey is between 0.37 and 0.55. The probability of detection is the fraction of the population that is observed during a flight within the area covered by the aircraft [33]. In this study, we use the same sightability probability as that used in previous aerial surveys conducted in Baja California (0.60), but with the assumption that the probability of sight can only refer to the animals seen and not to the total population.

The area sampled is the strip of land overflown by the aircraft during the survey hours, with a width of 250 m on each side of the flight path (500 m total), because this was considered the maximum distance at which animals can be sighted. To create this strip, we used the helicopter’s GPS track line to create a buffer area of 250 m on either side of the aircraft’s route. We determined the total area of habitat available to the species using the potential distribution model for bighorn sheep developed by Gutiérrez et al. [22]. We transformed the continuous probability presence model into a presence–absence model with a threshold of 0.65 [36]. We used QGIS (3.22.10) software for the management of geospatial information [37].

The number of animals observed, sightability, area sampled, and total area available for the species were integrated into the following formulas to estimate the bighorn sheep population based on an aerial survey:na=np
where *n_a_* is the estimated population of sheep in the sampled area, *n* is the number of sheep observed in the sampled area, and *p* is the sightability percent.

We then developed a formula that allowed us to extrapolate the result of *n_a_* via the total area of available habitat (*A*), resulting in the estimated total population of bighorn sheep for the entire distributional range (*N_A_*): NA=naaA=na.Aa
where *a* is the area sampled and *A* = is the total area of available habitat.

Statistical comparisons between hours of flight, observation rates, and number of sheep observed between the 2010 and 2021 surveys were analyzed with the nonparametric Mann–Whitney and Kruskal–Wallis tests. IBM SPSS statistics v.2.2 software was used for this statistical analysis [38].

## 3. Results

In 30.5 h of flight time, we surveyed 178,953 ha in total (Table 2), representing 40% of the potential bighorn sheep habitat in the state of Baja California (449,987 ha). We observed 456 bighorn sheep: 156 rams (29 class I, 24 class II, 39 class III, and 64 class IV rams), 252 adult ewes, 25 lambs, and 23 yearlings (14 rams and 9 ewes). The observation rate was 16 sheep sighted per hour of flight, and the ram:ewe:lamb:yearling ratio was 62:100:10:9, respectively, with an estimated 1697 ± 80 bighorn sheep in the state of Baja California.

We recorded the highest number of bighorn sheep and the highest observation rates in the Agua de Soda (83 sheep observed; 52 sheep observed per hour of flight) and Las Pintas (53 sheep observed; 23 sheep observed per hour of flight) mountain ranges (Table 2). The mountain ranges with the highest estimated populations of the species were Santa Isabel (361 sheep), La Libertad (255 sheep), and Sierra Juárez (214). On the other hand, Calamajué, Cucapá, Las Tinajas, and San Francisquito had the lowest observation rates of the species recorded, and in none of these mountain ranges did the estimated population exceed 50 individuals.

The results of the 2021 aerial monitoring can be compared with the results of the four previous surveys to evaluate changes in bighorn sheep populations in the northern and central mountain ranges of Baja California, but only by comparing the 2010 and 2021 surveys can an evaluation of changes in the population of the species statewide be made, since these are the only two studies in which all bighorn sheep distribution areas were surveyed (Table 3). In this sense, there was no difference in flight hours (pooled *t* tests, *p* = 0.96), number of sheep observed (Mann–Whitney U, *p* = 0.96), and observation rate (pooled *t* tests, *p* = 0.96) between the 2010 and 2021 surveys. Similarly, there was no difference in the number of sheep observed (Kruskal–Wallis test, *p =* 0.74) and the observation rate (one-way ANOVA, *p =* 0.75) in the northern and central mountain ranges of Baja California between the 1992, 1995, 1999, 2010, and 2021 aerial surveys.

Based on the results of the five aerial surveys conducted in Baja California, it was determined that the mountain ranges in northern and central Baja California with the highest average number of sheep observed and the highest observation rates were Santa Isabel and San Felipe. However, statistical analyses showed that there was no difference in the number of sheep observed (Table 4) and the observation rate (Table 5) between the northern and central mountain ranges of Baja California. Except for the Cucapá mountains, the number of sheep observed was lower than that registered in the Santa Isabel and San Felipe mountains and the observation rate was lower than recorded in the Santa Isabel mountains.

The 2021 survey recorded the highest number of rams per 100 ewes compared to the 2010 aerial survey, but also recorded the lowest number of yearlings and lambs per 100 females (Table 6). In 2010, the ratio of rams per 100 ewes was 51:100, and in 2021, it was 62:100. The number of yearlings and lambs per 100 ewes was no less than 50 in 2010 survey, but the ratio was 19:100 in 2021.

We recorded 114 sightings, of which 32 (28%) were singletons, 25 (22%) were in pairs, and 57 (50%) were sheep in groups of ≥3. Of the singletons, 25 (78%) were rams and 7 (22%) were ewes. Of the pairs we observed, nine (36%) were ewes, five (20%) were rams, five (20%) were rams and ewes, four (16%) were ewes and lambs, and two (8%) comprised a ewe and yearling. The mean group size of herds ≥ 3 was seven sheep, with groups ranging from three to twenty-three animals. Of these, 20 (35%) of the groups consisted of adults of both sexes, subadults, and lambs, 14 (25%) were of ewes and lambs, 13 (23%) were of rams and ewes, 7 (12%) were ewes only, and 3 (5%) were of rams only.

In Sierra Juárez and San Pedro Mártir, sightings of sheep occurred in the central and northern areas of the mountain ranges; in Cucapá and San Felipe, animals were observed in the northern areas; in Santa Isabel and La Asamblea, animals were observed in the drainage basins of the Gulf of California; in San Francisquito, animals were seen in the southern portion of the mountain range; in Las Tinajas, Las Pintas, La Libertad, Las Ánimas, and Agua de Soda, the sheep were sighted scattered throughout the mountain ranges (Figure 2).

## 4. Discussion

A comparison between the results of the number of sheep observed and the observation rate of the 2010 vs. 2021 surveys conducted in Baja California suggests that the desert bighorn sheep population in the state has remained stable. The same is true for the populations in the northern and central mountain ranges of the state, which remained stable between 1992 and 2021. Based on this information, it is plausible to suggest that there has been no significant change in the Baja California bighorn sheep population in 29 years. This is consistent with the conclusion of Lee et al. [20], who indicated that the population of Baja California bighorn sheep changed little between 1992 and 2010.

The stability of the Baja California bighorn sheep population contrasts with reports from Colorado, Nevada, and New Mexico, where species populations increased by 57%, 111%, and 425%, respectively, between 1992 and 2021 [39,40,41]. The increase in sheep populations in these states is probably due to the integrated management of the species, which includes surveillance, monitoring, the construction of guzzlers, translocations, and the eradication of feral fauna. This is in contrast to Baja California, where the management strategy consists only of maintaining the ban on sport hunting, and only a few self-motivated rural communities and non-governmental organizations conduct activities that contribute to the conservation of the species. Full legislative protection is a good-faith effort, but it does not address the factors that threaten bighorn sheep conservation, including illegal hunting, habitat degradation due to overgrazing by domestic livestock, climate change, and diseases [42].

The mountain ranges with the highest estimated numbers of bighorn sheep were Santa Isabel, La Libertad, and Sierra Juárez. The habitat available in these mountain ranges is extensive, according to the potential distribution model developed by Gutiérrez et al. [22]. Agua de Soda, Las Ánimas, and Las Pintas also supported that there were high populations of bighorn sheep (Table 2). In the 2010 survey, Las Ánimas, La Libertad, and Agua de Soda were also identified as mountain ranges with important populations of the species. On the other hand, Santa Isabel and Las Pintas had a low observation rate in the 2010 survey (Table 3). We attributed the higher observation rate in 2021 for these ranges to the more suitable habitats encompassed in the flight routes we followed.

There were also differences in the observation rates between the aerial surveys in 2010 and 2021 for the Cucapá, Las Tinajas, and La Asamblea mountain ranges (Table 3). In Cucapá and Las Tinajas, the observation rates were considerably greater in 2010 for unexplained reasons. In La Asamblea, a higher observation rate was recorded in 2021 [29] because although there were only four group sightings of sheep, one was of a group of twenty-three animals (Table 2), which was the largest group of sheep observed in this study.

The number of bighorn sheep observed in Cucapá in all the surveys conducted in Baja California was lower than the number of bighorn sheep observed in San Felipe and Santa Isabel (Table 4), which we attribute to the fact that the Cucapá mountain range had the fewest flight hours of observation during the surveys conducted in the state. On the other hand, the observation rate in Santa Isabel was higher than that in Cucapá, suggesting that the population size of bighorn sheep in Santa Isabel is larger than that in Cucapá. We consider this assertion plausible because Santa Isabel has a greater amount of available habitat (Table 3) and is farther from the large population centers of Baja California than it is from Cucapá (Figure 1). In addition, Cucapá is isolated from the rest of the Baja California mountain ranges by the Laguna Salada, which separates it from the Sierra Juárez by approximately 20 km at its narrowest point (Figure 1), so it is likely that the bighorn sheep population in Cucapá experiences periodic declines due to adverse environmental conditions in this isolated area.

The highest ram per 100 ewes ratios were recorded during the 2021 survey (62:100; Table 6). The ram:ewe ratio recorded in Baja California in 2021 was higher than that reported in Sonora (35:100) [43], a Mexican state where sport hunting is allowed, but similar to that reported in Arizona (53:100) [44] and Nevada (50:100) [40], U.S. states where sport hunting is also allowed. These results suggest that high ram:ewe rations can be maintained in wild sheep populations with sport hunting management programs.

The ratio of yearlings and lambs per 100 ewes in the 2021 survey is the lowest reported for the state (Table 6) and is also lower than that in Sonora (36:100) [43] and the desert mountains of Arizona (36:100) [44] and Texas (48:100) [45]. In desert bighorn sheep populations, the low proportions of yearlings and lambs is due to the fact that recruitment tends to occur in boom–bust cycles [12]. A similar low ratio of yearlings and lambs per 100 ewes found in this study (19:100) was reported in the Santa Rosa Mountain range in southern California in 1977 (11:100) [14], due to an epizootic (parainfluenza-3 and bluetongue) virus that caused high lamb mortality [46]. The lowest ratio of lambs and yearlings per 100 ewes in the history of desert bighorn sheep aerial surveys was recorded in Nevada in 2020 (21:100), and was due to high lamb mortality caused by an outbreak of pneumonia (*Mycoplasma ovipneumoniae*) and multi-year drought conditions across most of Nevada [38]. In the desert mountains of New Mexico and southern California, bighorn sheep populations declined based on recent surveys in New Mexico due to the presence of *Mycoplasma ovipneumoniae* [39] and in California due to a severe drought [47].

Most of the bighorn sheep recorded were observed in groups of between three and twenty-three animals. The most common groupings were of mixed-sex groups. In addition, a high number of sightings (25) were of solitary rams. These results indicate that the survey was conducted during the species’ mating season, as this is when rams join ewe groups searching for ewes in estrus [48,49].

## 5. Conclusions

Our data suggest stability in the bighorn sheep population in Baja California between 1992 and 2021. The mountain ranges that hosted the largest populations of the species in 2021 were Santa Isabel, La Libertad, Sierra Juárez, Agua de Soda, Las Ánimas, and Las Pintas. In the 2021 study, the low ratios of lambs and yearlings was probably due to a drought in 2021, but a disease event causing high mortality of lambs and yearlings cannot be ruled out. The study highlights the need to standardize wild sheep aerial surveys by defining flight paths and establishing a consistent duration of flights, and that flight surveys should be conducted by experienced monitoring personnel. Additionally, the state wild sheep management strategy has failed to achieve its primary goal of increasing the bighorn sheep population; it lacks concrete actions to benefit the conservation of the species. The state should prioritize initiating community-based wildlife conservation programs in the rural sector that are economically and ecologically effective in sustaining and enhancing wildlife biodiversity.

## Figures and Tables

**Figure 1 animals-14-00504-f001:**
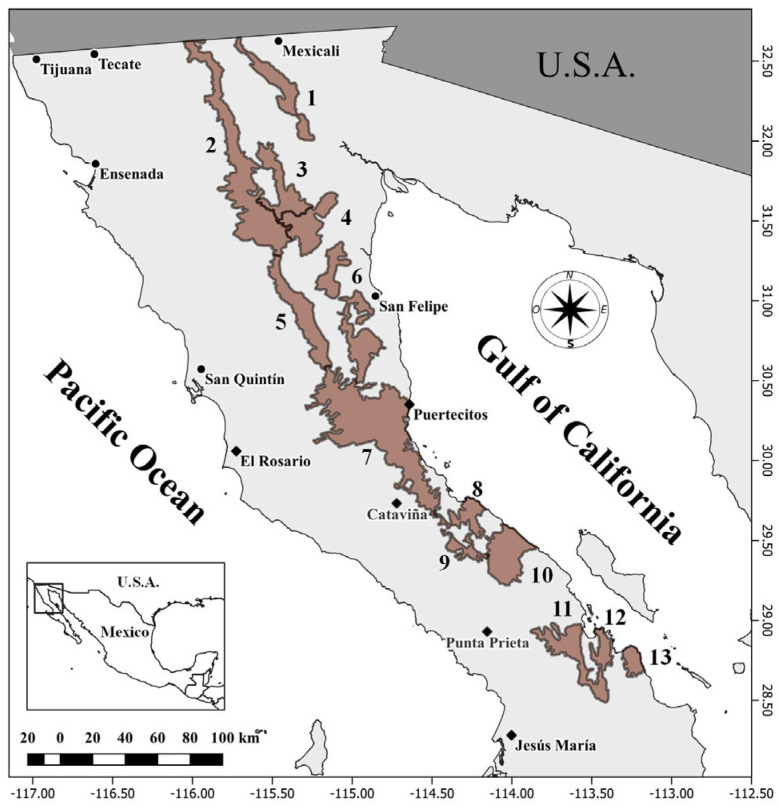
Sheep-populated mountain ranges of the state of Baja California, Mexico (brown polygons): (1) Cucapá; (2) Sierra Juárez; (3) Las Tinajas; (4) Las Pintas; (5) San Pedro Mártir; (6) San Felipe; (7) Santa Isabel; (8) San Francisquito; (9) Calamajué; (10) La Asamblea; (11) La Libertad; (12) Las Ánimas; (13) Agua de Soda. Cities are denoted by dots and towns by diamonds.

**Figure 2 animals-14-00504-f002:**
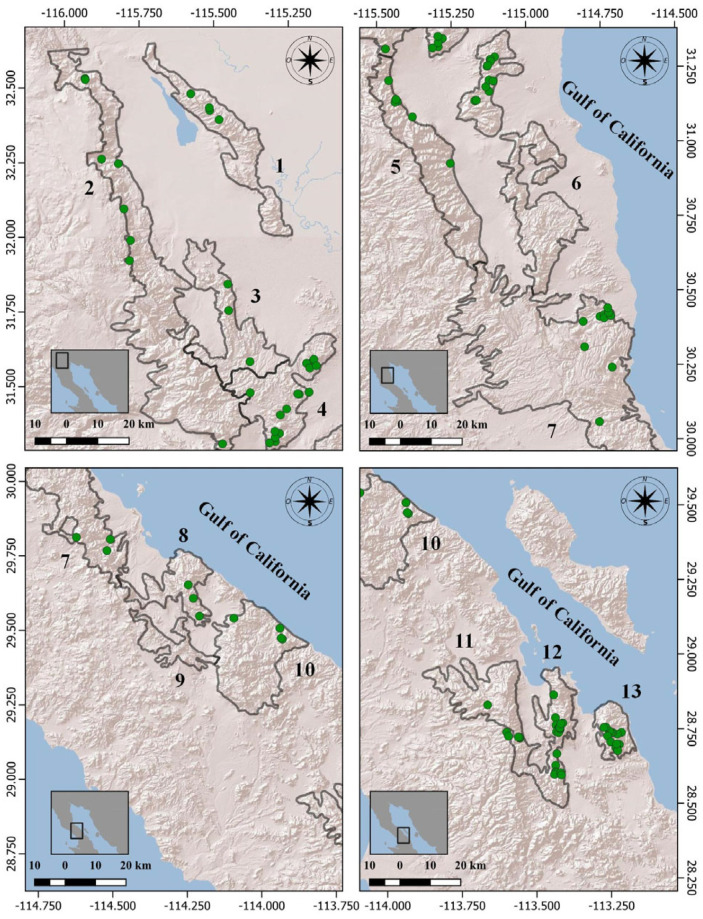
Location of sheep sightings during the 2021 aerial survey (green dots), in the state of Baja California, Mexico. The sheep-populated mountain ranges are as follows (black-outline polygons): (1) Cucapá; (2) Sierra Juárez; (3) Las Tinajas; (4) Las Pintas; (5) San Pedro Mártir; (6) San Felipe; (7) Santa Isabel; (8) San Francisquito; (9) Calamajué; (10) La Asamblea; (11) La Libertad; (12) Las Ánimas; (13) Agua de Soda.

**Table 1 animals-14-00504-t001:** Description of bighorn sheep age class and sex.

Age Class and Sex	Description
Lamb	Are no more than 1 m in height at the shoulder and have horns no longer than 125 mm.
Yearling ewe	Have slender, straight, and sharp pointed horns between 127 and 180 mm, and are larger than lambs (1.5 m shoulder height).
Ewe	Have horns of at least 125 mm in length each as measured on the outside curve of the horn from the skull to the tip.
Yearling ram	Appear similar to an adult ewe except for the face, which is shorter, and the nose resembles that of a lamb; the horns appear very much like the adult ewe horn but are thicker at the base and are blue-gray in contrast to the light-brown color of those of the adult ewe.
Class I ram	Have horns that are thick at the base and begin to curve downward; often have one prominent and one or two less prominent rings on the horn.
Class II ram	The horns are curved backward and downward to form a semicircle.
Class III ram	The horn tips are at eye level.
Class IV ram	The tips of the horns reach at least to the level of the eyes when broken and extend beyond this point when intact.

**Table 2 animals-14-00504-t002:** Summary of results of the aerial survey conducted in 2021 on the bighorn sheep population in the state of Baja California, Mexico.

Mountain Range	Available Habitat (ha)	Sampled Area (ha)	Sampling Time (h)	Rams	Ewes	Lambs	Yearlings		Number of Sightings	Sheep Observed	Estimated Population
							Rams	Ewes			
Cucapá	21,797	7014 (32 *)	0.8	4	1	1	0	0	4	6	31
Sierra Juárez	42,364	12,512 (30 *)	2.0	12	21	5	0	0	7	38	214
Las Tinajas	32,359	8069 (25 *)	1.0	2	2	0	0	0	3	4	27
Las Pintas	31,981	16,896 (53 *)	2.3	17	29	7	0	0	17	53	167
San Pedro Mártir	51,291	17,112 (33 *)	3.3	4	9	0	0	0	6	13	65
San Felipe	43,000	17,023 (40 *)	3.0	12	30	1	1	0	11	44	185
Santa Isabel	65,961	24,648 (37 *)	4.7	36	37	8	0	0	15	81	361
San Francisquito	18,373	11,228 (61 *)	1.8	4	0	0	0	0	3	4	11
Calamajué	17,616	4851 (28 *)	0.7	0	0	0	0	0	0	0	0
La Asamblea	57,158	25,656 (45 *)	2.9	12	12	1	3	2	4	30	111
La Libertad	42,721	15,631 (37 *)	4.9	13	35	2	3	3	13	56	255
Las Ánimas	17,088	10,987 (64 *)	1.5	21	21	0	1	1	15	44	114
Agua de Soda	8279	7326 (88 *)	1.6	19	55	0	6	3	16	83	156
Total	449,987	178,953 (40 *)	30.5	156	252	25	14	9	114	456	1697

* Percentage of sample area.

**Table 3 animals-14-00504-t003:** Comparison of the hours of flight, number of sheep observed, and observation rate of aerial surveys conducted in the state of Baja California, Mexico.

Mountain Range	Year														
	1992			1995			1999			2010			2021		
	Flight Hours	Sheep Obs.	Obs. Rate	Flight Hours	Sheep Obs.	Obs. Rate	Flight Hours	Sheep Obs.	Obs. Rate	Flight Hours	Sheep Obs.	Obs. Rate	Flight Hours	Sheep Obs.	Obs. Rate
Cucapá	2	2	1	0.5	0	0	----	----	----	1.5	19	13	0.8	6	8
Sierra Juárez	13	4	0.3	2.8	2	0.2	----	----	----	1.9	16	8	2.0	38	19
Las Tinajas	5.5	67	12	2.1	23	11	2	25	12	2	53	26	1.0	4	4
Las Pintas	4.5	25	6	2	27	13	1.2	1	1	2.3	22	10	2.3	53	23
San Pedro Mártir	14	83	6	6.7	14	3	3.7	72	20	4.7	16	3	3.3	13	4
San Felipe	18	282	16	5.1	85	5	5	25	5	5.4	50	9	3.0	44	15
Santa Isabel	11	140	13	5.3	111	21	4.2	125	30	3.9	20	5	4.7	81	17
San Francisquito	----	----	----	----	----	----	----	----	----	2	0	0	1.8	4	2
Calamajué	----	----	----	----	----	----	----	----	----	----	----	----	0.7	0	0
La Asamblea	----	----	----	----	----	----	----	----	----	2.4	7	3	2.9	30	10
La Libertad	----	----	----	----	----	----	----	----	----	2.4	77	32	4.9	56	11
Las Ánimas	----	----	----	----	----	----	----	----	----	1.2	58	48	1.5	44	29
Agua de Soda	----	----	----	----	----	----	----	----	----	1.5	43	29	1.6	83	52
Total	68	603	8	24.5	262	9	16.1	248	14	31.2	381	15	30.5	456	16

**Table 4 animals-14-00504-t004:** Results of the Kruskal–Wallis test comparing the number of sheep observed in the mountain ranges in northern and central Baja California based on an aerial survey conducted in 1992, 1995, 1999, 2010, and 2021.

Mountain Range	N	Sum of Scores	Expected under H0	Std Dev under H0	Mean Score	*p*-Value
Cucapá	4	23.50	68.0	18.11	5.87	0.016
Sierra Juárez	4	39.50	68.0	18.11	9.87	
Las Tinajas	5	86.00	85.0	19.90	17.20	
Las Pintas	5	75.50	85.0	19.90	15.10	
San Pedro Mártir	5	81.50	85.0	19.90	16.30	
San Felipe	5	122.00	85.0	19.90	24.40	
Santa Isabel	5	133.00	85.0	19.90	26.60	

**Table 5 animals-14-00504-t005:** Results of one-way ANOVA and Scheffe multiple comparison comparing observation rate (df = 6.26; F = 1.30; *p*-value = 0.29) between Cucapá (C), Sierra Juárez (SJ), Las Tinajas (LT), Las Pintas (LP), San Pedro Mártir (SPM), San Felipe (SF), and Santa Isabel (SI) based on an aerial survey conducted in 1992, 1995, 1999, 2010, and 2021 in the state of Baja California, Mexico.

Observation Rate	C 5.5 ± 6.13	SJ 6.87 ± 8.87	LT 13.0 ± 8.0	LP 10.60 ± 8.26	SPM 7.2 ± 7.25	SF 10.0 ± 5.29	SI 17.20 ± 9.28
C 5.5 ± 6.13	1	0.8029	0.1590	0.3332	0.7450	0.3923	0.0323 *
SJ 6.87 ± 8.87	0.8029	1	0.2471	0.4779	0.9504	0.5510	0.0565
LT 13.0 ± 8.0	0.1590	0.2471	1	0.6268	0.2451	0.5438	0.3970
LP 10.60 ± 8.26	0.3332	0.4779	0.6268	1	0.4919	0.9030	0.1876
SPM 7.2 ± 7.25	0.7450	0.9504	0.2451	0.4919	1	0.5708	0.0505
SF 10.0 ± 5.29	0.3923	0.5510	0.5438	0.9030	0.5708	1	0.1518
SI 17.20 ± 9.28	0.0323 *	0.0565	0.3970	0.1876	0.0505	0.1518	1

* Significant pairwise difference (*p* ≤ 0.05).

**Table 6 animals-14-00504-t006:** Comparison of sex and age class ratios in aerial surveys conducted in the state of Baja California, Mexico.

Classification	1992	1995	1999	2010	2021
Ram	32	57	61	51	62
Ewe	100	100	100	100	100
Lamb	45	43	25	54	10
Yearling	22	9	27	20	9

## Data Availability

The data presented in this study are available on request from the corresponding author. The data are not publicly available due to it is owned by the Universidad Autónoma de Baja California.

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
