# Peer review of "Population and Conservation Status of Bighorn Sheep in the State of Baja California, Mexico"

_animals, 2024, doi:10.3390/ani14030504_

Round 1

Reviewer 1 Report

Comments and Suggestions for Authors

The introduction describes the problem quite well and provides the reader with all the necessary information.

Line 36/37 is it necessary to name the full title of the report. The reference should be enough

Line 61 not only big horn sheep monitoring – monitoring in general

Abundance is the number of animals in a given area

Age structure of the population was described and discussed I miss the same for sex ratio

Line 98-102 this should be shifted to study area

Line 106-110 previous surveys should be mentioned but not described in this detail in the introduction

Study area

Link should be shifted to the references

Information from Fig. 1 should be mentioned here

Arial survey

Start with the actual survey and described possible problems of missing information afterwards

Line 162-173 In my opinion this information is easier the understand when it is presented in a table

Line 188-191 probability of sight can only refer to the animals seen and not to the total population

Shift the links to the references

Line 204 the calculation is quite simple and should be clear to a reader with scientific background

This is the same for the formula in line 211

Description of table 2 – the years of the survey are presented in the table and it is not necessary to describe them in the description again. Is it possible to use acronyms to describe the factors and not A, B, C. Will make the table easier to understand

Line 234-246 Is it necessary to describe these results in detail. In may opinion a more general description to understand the general results would be enough.

Table 3 is not necessary. Results can be included in the text.

Table 4 it is not necessary to name all the mountain ranges and years in the description and in the table. In the table is enough. There is only one p value – because there is a significant difference? I do not find the result in the text.

Table 6 years in the description

Is it necessary to present the distribution of sightings 2021. The information was presented in the tables and Fig 2 does not present new information which is necessary to understand change in population size of structure and there is no comparison with distributions from former survey. The information I interesting from a faunistical standpoint but this is not the focus of the paper

Discussion

Line 300-302 in my option the numbers are not necessary

Shift links

It does not make sense to compare number of seen sheep when ii is linked with flight time. It is much more important to compare populations sizes or abundances the describe the importance of different mountain ranges for the population

Lin e342-348 the authors describe that a comparison between former surveys is not possible. How does this fit to figure 3 and how do you explain the different results from the different sites?

Discussion to population structure is okay for me

There is no analysis of the distribution. The assumption is logical and the discussion too but the pattern could also be a result of the chosen flight path.

Links to the references

Conclusion is okay.

Only one thought about stability of the population. Comparing the data from 2010 and 2021 (Table 2) flight time is quite similar 31,2 to 30,5 h but absolute number of counted animals was much higher in 2021 (381 to 456 sheep means 20% more). Is it possible that stability was detected mainly because of the high variation in the data Table 3 and is more a methodical problem than a reals biological result?

Author Response

Response to Reviewer 1 Comments

Point 1: Line 36/37 is it necessary to name the full title of the report. The reference should be enough. 

Response 1: At the request of one of the reviewers, we have modified the wording of the brief abstract and the abstract to more clearly present the results of the study.

Point 2: Line 61 not only big horn sheep monitoring – monitoring in general.

Response 2: We agree.

Point 3: Abundance is the number of animals in a given area.

Response 3: The word "location" was changed to "area" (line 69).

Point 4: Age structure of the population was described and discussed I miss the same for sex ratio.

Response 4: Attended (line 83-88). ¨On the other hand, the sex ratio is a parameter that must be carefully monitored in populations exploited by sport hunting, since the decrease in the proportion of males is mainly related to overharvest and, therefore, when it is detected, measures must be taken to reduce the pressure on the populations [14]. Unstable age structures and sex ratios can have genetic or reproductive consequences.¨

Point 5: Line 98-102 this should be shifted to study area. 

Response 5: We have changed the wording of the paragraph to clarify the information (line 96-107). 

Point 6: Line 106-110 previous surveys should be mentioned but not described in this detail in the introduction.

Response 6: modified paragraph (Line 108-115): ¨Four aerial surveys have been conducted in Baja California to assess the abundance and population structure of bighorn sheep: 1992, 1995, 1999, and 2010. The 1992, 1995, and 1999 surveys covered half of Baja California's mountain ranges where the species occurs [26-28]; only the 2010 survey covered all of the state's mountain ranges where bighorn sheep occur [29]. The aims of this study were to estimate the current abundance and population structure of bighorn sheep populations in the state of Baja California based on the aerial survey conducted in 2010 and to compare the 2021 data with previous surveys to establish historical population size trends.

Point 7: Study area. Link should be shifted to the references.

Response 7: We shifted the links to the references.

Point 8: Information from Fig. 1 should be mentioned here.

Response 8: The information in Figure 1 can be found on lines 118-124.

Point 9: Aerial survey. Start with the actual survey and describe possible problems of missing information afterwards. 

Response 9: Attended. We changed the order of the narrative to leave the problems of aerial surveys in Baja California at the end. (line 137-158).

Point 10: Line 162-173 In my opinion this information is easier the understand when it is presented in a table.

Response 10:  We have prepared a table (1)

Point 11: Line 188-191 probability of sight can only refer to the animals seen and not to the total population.

Response 11: We modify the text (line 170-171).

Point 12: Shift the links to the references.

Response 12: We shifted the links to the references.

Point 13: Line 204 the calculation is quite simple and should be clear to a reader with scientific background. 

Response 13: We believe that the equation is very understandable to any reader with a scientific background.

Point 14: This is the same for the formula in line 211.

Response 14: We believe that the equation is very understandable to any reader with a scientific background.

Point 15: Description of table 2 – the years of the survey are presented in the table and it is not necessary to describe them in the description again. Is it possible to use acronyms to describe the factors and not A, B, C. Will make the table easier to understand.

Response 15: We modify the description of the table and change A, B and C by description of the factors.

Point 16: Line 234-246 Is it necessary to describe these results in detail. In my opinion a more general description to understand the general results would be enough.

Response 16: Change the wording of the paragraph that introduces Table 3 (line 214-224).

Point 17: Table 3 is not necessary. Results can be included in the text. 

Response 17: We remove the table and include the information in the text (line 219-224).

Point 18: Table 4 it is not necessary to name all the mountain ranges and years in the description and in the table. In the table is enough. There is only one p value – because there is a significant difference? I do not find the result in the text.

Response 18: Modify the title of the table; This P-value (0.016) is the general P-value for the Kruskal-Wallis significance test. Since the P-value is smaller than 0.05, we conclude that at least there is a difference among the sum of scores of the mountain range. We compared the sum of scores using the Dunn test and found significant differences in the number of animals for Santa Isabel and San Felipe vs Cucapá; This information can be found on lines 230-234.

Point 19: Table 6 years in the description.

Response 19: We modify the description on the table.

Point 20: Is it necessary to present the distribution of sightings 2021. The information was presented in the tables and Fig 2 does not present new information which is necessary to understand change in population size of structure and there is no comparison with distributions from former survey. The information I interesting from a faunistical standpoint but this is not the focus of the paper.

Response 20: We consider that Figure 2 should be kept as it shows the points of agglomeration of bighorn sheep in the different mountain ranges of Baja California and this information is necessary to standardize the aerial monitoring of the species. This, according to our conclusions, is something that must be done to ensure the reliability of the results of aerial monitoring.

Point 21: Discussion. Line 300-302 in my option the numbers are not necessary. 

Response 21: We have changed the wording of these lines to change the numbers to percentages. With the new wording, we believe the reading is more agile and the message is stronger (line 280-282).

Point 22: Shift links.

Response 22: We shifted the links to the references.

Point 23: It does not make sense to compare number of seen sheep when is linked with flight time. It is much more important to compare populations sizes or abundances the describe the importance of different mountain ranges for the population.

Response 23: The comparison of bighorn sheep population abundance can be found in lines 214-224 and 272-279. On the other hand, we believe it is important to maintain the  comparison between mountain ranges because it is critical information for the management of the species and this is the first paper to present an in-depth analysis on this topic.    

Point 24: Line 342-348 the authors describe that a comparison between former surveys is not possible. How does this fit to figure 3 and how do you explain the different results from the different sites?

Response 24: We removed figure 3; it is not possible to compare years because of the differences in surveys protocols as stated previously in the text.

Point 25: Discussion to population structure is okay for me. 

Response 25: Ok.

Point 26: There is no analysis of the distribution. The assumption is logical and the discussion too but the pattern could also be a result of the chosen flight path.

Response 26: Okay, we will remove that paragraph from the discussion.

Point 27: Links to the references.

Response 27: We shifted the links to the references.

Point 28: Conclusion is okay.

Response 28: Ok.

Point 29: Only one thought about stability of the population. Comparing the data from 2010 and 2021 (Table 2) flight time is quite similar 31,2 to 30,5 h but absolute number of counted animals was much higher in 2021 (381 to 456 sheep means 20% more). Is it possible that stability was detected mainly because of the high variation in the data Table 3 and is more a methodical problem than a real biological result?

Response 29: Yes, the variation in Table 3 is a methodical problem and data between years cannot be compared.  We removed Table 3.  Refer to response 24.

Reviewer 2 Report

Comments and Suggestions for Authors

General Comments

This paper investigates the population and conservation status of bighorn sheep in the state of Baja California, Mexico. The paper recommends a shift toward community-based wildlife conservation programs, proven to be ecologically and economically effective in other Mexican states. The research underscores the importance of standardizing aerial surveys and advocates for a modified conservation approach to enhance bighorn sheep populations in Baja California. However, certain areas should be addressed to enhance the manuscript's potential for publication.

Specific comments:

Point 1: The author extensively used underlining throughout the entire manuscript. Could you please clarify the reasons behind using underlines for certain words? Kindly explain; otherwise, consider removing the underlines from the text.

Point 2: The simple summary and abstract appear to convey similar meanings, and in the abstract, the author did not explicitly outline the study's results. Please adhere to the journal's guidelines when crafting an abstract and ensure a clear presentation of the study's findings.

Point 3: While citing in the text, the author used pp. xxx-xxx. There is no need to mention it; please remove it from the text.

Point 4: On page 3, combine lines 106 to 115 into a single paragraph for better flow and coherence.

Point 5: On page 4, line 143, in section 2.2, please provide an appropriate subtitle for the paragraph.

Point 6: On page 9, line 294, could you clarify the intended meaning behind the use of that particular line? Please revise the line in a more meaningful manner to enhance clarity and coherence in the text.

Point 7: The images in Figure 3 are unclear. Please use high-resolution images for Figure 3 to enhance clarity.

Point 8: The presentation of the results in the manuscript is confusing. Please present the results clearly and concisely to enhance reader understanding.

Author Response

Response to Reviewer 2 Comments

Point 1: The author extensively used underlining throughout the entire manuscript. Could you please clarify the reasons behind using underlines for certain words? Kindly explain; otherwise, consider removing the underlines from the text. 

Response 1: We remove the underline from the text.

Point 2: The simple summary and abstract appear to convey similar meanings, and in the abstract, the author did not explicitly outline the study's results. Please adhere to the journal's guidelines when crafting an abstract and ensure a clear presentation of the study's findings.

Response 2: We have modified the wording of the simple summary and abstract to more clearly present the study results.

Point 3: While citing in the text, the author used pp. xxx-xxx. There is no need to mention it; please remove it from the text.

Response 3: We removed the number of pages on the text.

Point 4: On page 3, combine lines 106 to 115 into a single paragraph for better flow and coherence.

Response 4: We combined the paragraphs.

Point 5: On page 4, line 143, in section 2.2, please provide an appropriate subtitle for the paragraph.

Response 5: We provide an appropriate subtitle.

Point 6: On page 9, line 294, could you clarify the intended meaning behind the use of that particular line? Please revise the line in a more meaningful manner to enhance clarity and coherence in the text.

Response 6: We have corrected the mistake.

Point 7: The images in Figure 3 are unclear. Please use high-resolution images for Figure 3 to enhance clarity.

Response 7: Improving the resolution of Figure 3.

Point 8: The presentation of the results in the manuscript is confusing. Please present the results clearly and concisely to enhance reader understanding.

Response 8: We have changed the wording of the results to make them easier to read and understand.

Reviewer 3 Report

Comments and Suggestions for Authors

Dear Authors,

make changes according to the comments.

Author Response

Response to Reviewer 3 Comments

Point 1: Line 32. A week of research is extremely short to assess the state of the population. The studies were carried out only in the autumn. For a comprehensive assessment, comparative data from research results in other seasons of the year are needed. If they exist, please provide them. 

Response 1: We  followed the established methodologies for monitoring bighorn sheep throughout North America.  Comparative seasonal data is not available. 

Point 2: Line 34. Based on 30 hours of observations, you identify the population size and extrapolate it to the entire area, increasing it fourfold. This is a fairly rough calculation. I repeat, over such a short period of one week, it is difficult to assess the state of the population, much less draw conclusions.

Response 2: We thank you for your observations, however, we followed the standard wildlife sheep survey methodology (line 137-149)

Point 3: Line 49. It is necessary to provide data on the number of hunting licenses issued per year of observation in the study area. The total number of animals killed in that year and the separate number of animals killed by mountain range hunters should be indicated. Calculate the percentage of hunting as a percentage of the total estimated population size. Recommend an acceptable percentage of sheep removal per year based on your results.

Response 3: Sheep hunting is prohibited in the state of Baja California, and has been since 1990 (line 56-57). Hence, data on hunting is not available. Recommending a percentage of sheep removal is not appropriate for our paper; however, we conclude that the strategy of prohibiting the harvesting of the species should be changed and initiatives should be developed that provide economic benefits to the communities that possess the resource (line 356-359).

Point 4: Line 155. The climatic conditions during the observation period should be given (temperature, precipitation, wind force, etc.).

Response 4: We did not record this information. The pilot determines if weather conditions are appropriate for flight procedures. This information is not reported in monitoring surveys.  If the pilot determines that flight conditions are favorable then weather conditions are also appropriate for sheep monitoring.

Point 5: Line 158. Why didn't you record the initial direction of movement of the sheep? This is important from the standpoint of understanding the migration routes of animals. Since you conducted research over the course of a week, do you rule out the possibility of recording the same individuals? This explanation must be added to the text of the manuscript. 

Response 5: The data on the movement of the sheep is not reported in surveys because they are biased; movements are usually affected by the presence of the helicopter.  In some instances sheep remain still. Hence this information is not recorded. We rule out the possibility of recording the same individuals because the helicopter covers a route only once. Once sheep are observed, the appropriate data is recorded and the helicopter assumes its flight path and does not fly over the same area twice.

Point 6: Line 213. It is known that animals of different species have the greatest daily activity at certain hours. How was this taken into account when conducting flight surveys? At what hours were the largest number of individuals recorded? Based on your results, this should be added to the method section and results of the manuscript.  

Response 6: Population surveys are conducted during the mating season when females and rams are in the same herds and hence both are readily visible.  Also, sheep avoid tall, dense vegetation so are visible throughout the day. On the other hand, hours of sheep activity are not a factor considered when scheduling aerial monitoring because low-altitude flights cause animals to move regardless of activity (Krausman and Hervert, 1983); therefore, flights are generally scheduled to take advantage of daylight hours. 

Krausman, P.; Hervert, J. Mountain Sheep Responses to Aerial Surveys. The Wildlife Society Bulletin, 1983, 11, 372-375. https://www.jstor.org/stable/3781675. 

Point 7: Line 218. A table should be added on the distribution of the number of observation hours for individual territories (mountain ranges). It is completely unclear whether the observation hours were evenly distributed across individual territories. This needs to be improved. Based on Table 1, comparing the volume of observation hours with the area of individual territories, an uneven distribution was revealed. If this is really true? It is necessary to explain why certain territories, based on a comparable area, received more observation hours, while other territories received fewer observation hours? 

Response 7: See table 2 column 4 (sampling time) which records the observation hours for each mountain range.  The number of flight hours on mountain ranges depended on the area of the mountain ranges. All observations are proportional to the size of the area of the mountain range. Hence some mountains receive more observation hours; nevertheless, in the conclusion we pointed out that as part of the standardization of flights in Baja California, the monitoring time to be used in each mountain range should be established (line 352-354). 

Point 8: Table 2. Based on these results, we should talk about normal primary, secondary and tertiary sex ratios in populations of this species. In addition, it is necessary to mention what cyclicity is known for this species and similar mammalian species? In which years are there population outbursts and when are population depressions observed?

Response 8: We followed the established procedure for recording sex and age ratios.  Cyclicity is not a term used in sheep biology. If by cyclicity you are referring to population dynamics, this aspect of sheep biology has been reported in several studies. This data is based on long-term studies and is not reported in sheep monitoring surveys.

Point 9: Line 260. A high proportion of males in animal populations usually indicates an optimal population size that is in equilibrium. And based on your results, you conclude that the population is decreasing in size. This is a question of extrapolation and possible subjectivity in estimating population density based on a week of observations only in the autumn. 

Response 9:  Agreed, the correct conclusion is the following: “Comparison of the results of the number of sheep observed and the observation rate of the 2010 vs 2021 surveys conducted in Baja California suggests that the desert bighorn sheep population in the state has remained stable” (line 272-274).

Point 10: Line 280-285. Here you should indicate the direction of movement of groups of animals. It is possible to indicate them with arrows on maps. 

Response 10: The data is not appropriate in this type of survey. The sound of the helicopter can erratically disperse the sheep.

Point 11: Line 294-302. This is quite strange, given the results presented on the sex structure of the population. One should not make such categorical premature statements based on the volume of observations.

Response 11: Agreed, we deleted the word “decreased” and substituted the words “has remained stable” (line 274).

Point 12: Line 342-348. This is where you contradict yourself.  ??

Response 12: We removed figure 3; it is not possible to compare years because of the differences in surveys protocols as stated previously in the text.

Point 13: Line 354-357. This is debatable. The ratio of males and females in populations is independently stable in the population structure. 

Response 13: Agreed, the increase in the proportion of males in the population may be due to multiple factors. We deleted the following sentence: “This increase in the proportion of rams may be due to the fact that bighorn sheep populations in Baja California have not been subjected to sport hunting pressure since 1990.”

Point 14: Line 369-375. Detailed similar data should be provided for the territory you studied.

Response 14: There are no recorded die-offs for sheep populations in Mexico due to droughts or disease. Hence we added the data from other studies in the USA as possible reasons for high mortalities.

Point 15: Line 392-394. This cannot be called a conclusion. This is more like speculation. Your manuscript does not mention the number of cases of recorded deaths of rams from disease. You should collect statistics on sheep diseases from human surveys. It is necessary to list the frequency of recorded disease outbreaks in this species based on long-term observations.

Response 15: As mentioned above there are no recorded instances of disease die-offs in Mexico.  There is a large literature on this subject for sheep populations in the USA. The inclusion of a review of disease  studies is not relevant to this study.

Point 16: Line 394-397. Your results do not provide clear, reasoned recommendations on this matter. If you want to recommend certain flight trajectories, then you need to write them clearly in the conclusions.

Response 16: We are not recommending specific flight procedures. We are recommending that biologists need to establish standard procedures based on a consensus as to what flight procedures should be followed to improve all future monitoring flights.

Point 17: Line 397-399. This is a controversial position based on the amount of research you have done on watches.  

Response 17:  We believe that this conclusion is quite accurate, as bighorn sheep populations cannot be expected to increase if the conservation strategy for the species consists solely of prohibiting sport hunting. As Whiting et al. (2023) point out, “Full legislative protection is a good-faith effort, but it does not address the factors that threaten bighorn sheep conservation, as including, illegal hunting, habitat degradation due to overgrazing by domestic livestock, climate change, and diseases.”

Whiting, J.; Bleich, V.; Bowyer, R.; Epps, C. Restoration of bighorn sheep: History, successes, and remaining conservation issues. Front. Ecol. Evol. 2023, 11. https://doi.org/10.3389/fevo.2023.1083350

Round 2

Reviewer 1 Report

Comments and Suggestions for Authors

Introduction

Line 54/55 delete NOM-059-2010

Line 58 could be a reference too - State strategy for the conservation and sustainable 58 management of the bighorn sheep (Ovis canadensis cremnobates) in Baja California

Line 70 stable, increasing, decreasing

Material, methods

Line 181-192 The calculations are quite simple and should be clear for a reader with scientific background. I recommend to skip this part.

Results

Table 2 - * should be indicated in each row

Line 228-234 – do these data revere to all surveys or only to 2010 and 2021?

I a, fine with Discussion and conclusions

Author Response

Point 1: Line 54/55 delete NOM-059-2010

Response 1: We deleted NOM-059-2010 (line 54).

Point 2: Line 58 could be a reference too - State strategy for the conservation and sustainable management of the bighorn sheep (Ovis canadensis cremnobates) in Baja California.

Response 2: We deleted “State strategy for the conservation and sustainable management of the bighorn sheep (Ovis canadensis cremnobates) in Baja California” and kept the reference number 7 on the text.

Point 3: Line 70 stable, increasing, decreasing.

Response 3: We made the suggested change (line 69-70).

Point 4: Line 181-192 The calculations are quite simple and should be clear for a reader with scientific background. I recommend to skip this part.

Response 4: We prefer to leave this part for clarity and because it is one of the main focuses of the survey.

Point 5: Table 2 - * should be indicated in each row

Response 5: We place the * in each row. 

Point 6: Line 228-234 – do these data revere to all surveys or only to 2010 and 2021?

Response 6: In all surveys (line 227-228).

Reviewer 3 Report

Comments and Suggestions for Authors

Dear Authors,

I am satisfied with the revised manuscript. Thank you.

Author Response

Response to Reviewer 3 Comments

We appreciate the time you took to review our work. Your comments were of great help in improving the quality of the manuscript. Greetings and best wishes for the New Year.